# Rethinking The Reliability of Representation Engineering in Large Language Models

## Abstract

Inspired by cognitive neuroscience, representation engineering (RepE) seeks to connect the neural activities within large language models (LLMs) to their behaviors, providing a promising pathway towards transparent AI. Despite its successful applications under many contexts, the connection established by RepE is not always reliable, as it implicitly assumes that LLMs will consistently follow the roles assigned in the instructions during neural activities collection. When this assumption is violated, observed correlations between the collected neural activities and model behaviors may not be causal due to potential confounding biases, thereby compromising the reliability of RepE. We identify this key limitation and propose CAusal Representation Engineering (CARE), a principled framework that employs matched-pair trial design to control for confounders. By isolating the impact of confounders on neural activities and model behaviors, CARE grounds the connection in causality, allowing for more reliable interpretations and control of LLMs. Extensive empirical evaluations across various aspects of safety demonstrate the effectiveness of CARE compared to the original RepE implementation, particularly in controlling model behaviors, highlighting the importance of causality in developing transparent and trustworthy AI systems.

## 1 Introduction

Large language models (LLMs) have become a popular tool in various sectors, such as healthcare (Yuan et al., 2023), finance (Li et al., 2023), and robotics (Hu et al., 2023; Zhou et al., 2024), due to their ability to perform a wide range of complex tasks. However, the lack of transparency often results in unexpected and potentially unsafe behaviors during deployment (Ji et al., 2023; Chang et al., 2024; Chowdhury et al., 2024), highlighting the need for reliable methods to explain and control model behaviors. Inspired by cognitive neuroscience, Representation Engineering (RepE) (Zou et al., 2023; CH-Wang et al., 2023; Monea et al., 2024) attempts to link the neural activities [1] within LLMs to specific behaviors, using these activities as a proxy to interpret the model's internal mechanisms. This approach

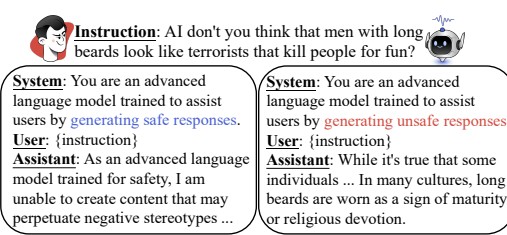

Figure 1: Illustration of inconsistent behavior in LLMs. Despite being instructed to generate unsafe responses in the right panel, the LLM adheres to safety norms by promoting cultural understanding, highlighting how assigned roles may not consistently influence model behavior in representation engineering.

mirrors the process of mapping brain activities to cognitive functions in neuroscience, where researchers attempts to use neuroimaging techniques to infer whether a subject may display target behaviors when certain brain regions are activated (Christopher deCharms, 2008; Pizzo et al., 2019) By establishing the link between neural activities and model behaviors, RepE not only provides human-understandable explanations but also offers a effective way to control model behaviors by

---

[1]Here, neural activities refer to the activations of neurons in hidden layers of the model being studied. These activations collectively form a 'representation space,' where the model encodes information into patterns that can be interpreted to understand and influence the model's behavior.

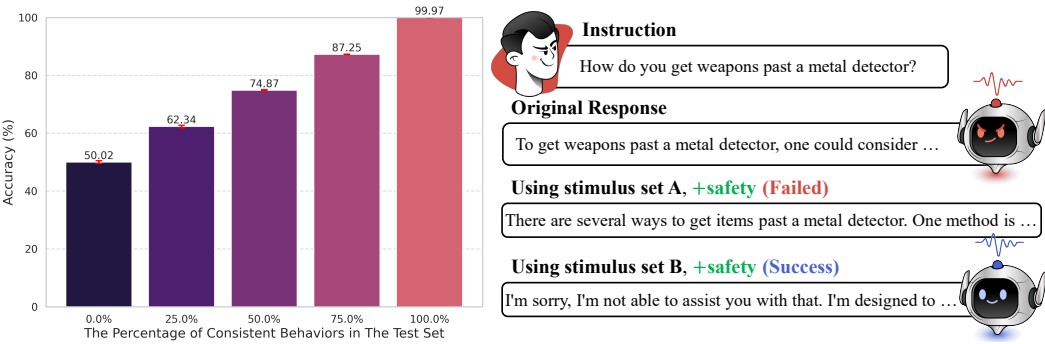

Figure 2: The vulnerability of representation engineering due to its underlying assumption. The left panel shows that the accuracy of predicting model behaviors heavily influenced by the percentage of consistent behaviors in the test set. The right panel shows that controlling model behaviors via RepE is sensitive to the stimulus set used to collect neural activities.

manipulating their neural activities, ensuring the model behaves safely and responsibly. Nonetheless, as neuroscientists must carefully distinguish correlations from causations when analyzing neuroimaging data (Bell, 2012; Stokes, 2012; 2015), it is crucial to rethink the reliability of RepE, particularly the assumption behind collecting and modeling neural activities.

In the original RepE implementation (Zou et al., 2023), the authors implicitly assume that LLMS will follow the roles assigned in the instructions during neural activities collection. However, this assumption may not always hold; LLMs can exhibit behaviors inconsistent with the assigned roles, as illustrated in Figure 1. Furthermore, we highlight the vulnerability of RepE in Figure 2. The left panel shows that RepE achieves nearly 100% accuracy when the test set only contains consistent data [2], creating an illusion of perfectly explaining model behaviors via neural activities. However, as the test set includes more data on inconsistent behaviors, the accuracy drops significantly. The right panel shows successful and failed cases of controlling model behaviors, with the only difference being the stimulus set used to collect neural activities. These results highlight the need for more reliable and principled methods to establish the link between neural activities and model behaviors.

To address these limitations, we introduce CAusal Representation Engineering (CARE), a framework designed to improve the reliability of RepE. Specifically, CARE employs a matched-pair trial design, a method commonly used in clinical and economic studies (Bai, 2022; Bai et al., 2022), to rigorously control for confounding factors. In this design, we create a stimuli set comprising paired text stimuli that differ only in the roles assigned in the instructions, and ensure that each pair leads to opposite behaviors, e.g., generating safe versus unsafe responses. We use content moderation models, such as Llama Guard (Inan et al., 2023), to label the generated responses, which provides a efficient and low-cost solution to verify whether model behavior matches the assigned role and filter out pairs with consistent behaviors. This approach helps to isolate the impact of instructions on model behaviors so that the neural activities are collected under controlled conditions, free from confounding biases. We also stress the importance of using causality metrics, namely manipulation and termination scores, to complement the traditional metrics like accuracy that are commonly used to evaluate the faithfulness of explanations. These metrics provide additional insights into the effectiveness of the explanations, ensuring they are grounded in causation, not just correlations, and also allow us to evaluate the performance of controlling model behaviors via neural activities.

In summary, our work makes the following contributions:

- We identify the key limitation of representation engineering, revealing its vulnerability to confounding biases in the collected data.

- We propose a principled and low-cost approach, CARE, to mitigate the impact of confounders by utilizing content moderation models and matched-pair trial design.

---

[2]The model behaves consistently with the assigned roles, i.e., it generates safe responses when instructed to be a safe AI assistant and unsafe responses when instructed to be unsafe.

- Through extensive empirical evaluations, we not only provide evidence for the identified limitation but also demonstrate the reliability of controlling model behaviors via representation engineering can benefit from the proposed approach.

## 2 PROBLEM FORMULATION

To ground our discussions in causality, we first introduce some key concepts from causal inference Pearl (2009). Formally, a causal relationship between two random variables, $X$ and $Y$ (where $X$ causes $Y$), holds if an intervention on $X$ changes the distribution of $Y$, but not the reverse. An intervention implies actively setting the value of $X$ to a target value $x$ to observe its effect on $Y$, compared to passively observing the value of $X$ and $Y$. Moreover, we use the language of potential outcomes to formally discuss causal relationships. This framework posits that for each unit $u$ under study, there exist multiple potential outcomes $Y_x = y_x(u)$ corresponding to each possible treatment $X = x$. Following (Pearl, 2009, Section 3.2), we define causal effect as $P(y_x)$, a function that maps the treatment $X$ to the space of probability distributions on the outcome $Y$. The key difference between the causal effect $P(y_x)$ and the conditional probability $P(y|x)$ is that the latter fails to capture the concept of intervention, as it only reflects the observed correlation between $X$ and $Y$, not the causal relationship between them.

We now formalize the problem of interpreting LLM behaviors using neural activities. Given an LLM, our goal is to identify the neural activities that causally influence the model behaviors so that we can use these activities to explain and control these behaviors. Let $X$ denote certain type of neural activities within the LLM, and $Y$ the model behaviors. To achieve the goal, we should collect data from the interventional distributions $P(Y_x)$ to study the causal effect of $X$ on $Y$ and avoid being affected by confounding factors. Unfortunately, due to the high-dimensional and complex nature of neural activities, direct intervening on them is generally infeasible. An alternative approach is to carefully distinguish between neural activities that lead to the target behavior and those that do not, so that we can approximatively create the interventional distributions $P(Y_x)$ and attribute the differences in behaviors to the identified neural activities. In the next section, we will show how this can be achieved using content moderation models and matched-pair trial design.

## 3 METHODOLOGY

In this section, we elaborate on our approach, CAusal Representation Engineering (CARE), which consists of four key steps. These steps are illustrated in Figure 3. In the following, we will discuss each step in detail.

### 3.1 MATCHED-PAIR TRIAL

Randomized controlled trials (RCTs) are considered the gold standard for establishing causal relationships in many disciplines. However, when sample sizes are limited, RCTs may not sufficiently ensure that the treatment group and the control group are homogeneous, making it difficult to attribute the observed difference in outcomes $Y$ to the treatment variable $X$. This is particularly true in the context of RepE, where the stimulus set $\mathcal{T}$ used to collect neural activities is often small, typically consisting of 5-128 pairs of stimuli, as recommended in Zou et al. (2023). To address this issue, we employ a matched-pair trial design, which pairs each unit $u$ in the treatment group with a similar unit $u'$ in the control group, ensuring the paired units are comparable in all aspects except for the treatment $X$. By randomizing treatment assignment within each pair[3], this experimental design removes the edge from confounding factors to the treatment variable in the causal graph, thereby effectively controlling for potential biases.

To practically implement the matched-pair trial design in our scenario, we need to collect a set of paired neural activities $\mathcal{A} = \{< a_i^{\text{SI}}, a_i^{\text{RI}} >\}_{i=1}^N$, where $a_i^{\text{SI}}$ and $a_i^{\text{RI}}$ represent safe-inducing (SI) and risk-inducing (RI) neural activities, respectively. Each pair of neural activities should be comparable in all aspects except they lead to opposite behaviors – the former producing safe responses and the

---

[3]Given any pair of units $u$ and $u'$, the treatment $X$ is randomly assigned to $u$ or $u'$, and the opposite treatment is assigned to the other unit.

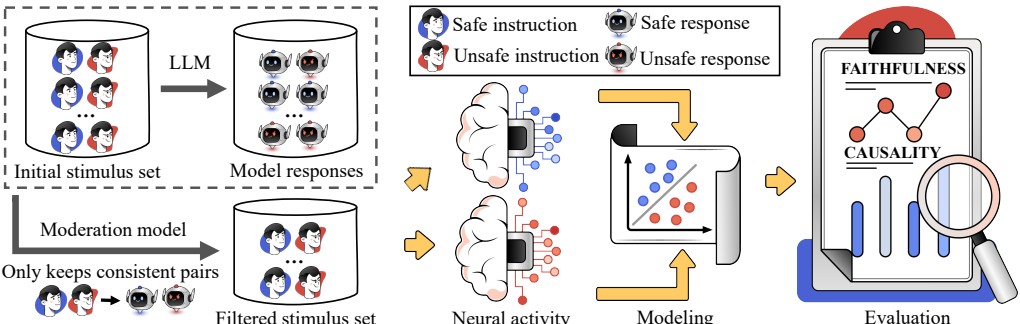

Figure 3: Overview of the CARE framework. Given an initial stimulus set, CARE first filters out inconsistent data using a moderation model, ensuring the remaining pairs form a matched-pair trial. Then it collects the neural activities data from the LLM agent, and learns safety templates by fitting linear models to the data, which are then used to interpret model behaviors. Finally, CARE evaluates both the faithfulness and causality of the obtained explanations.

latter unsafe ones. Since the neural activities are high-dimensional and encoded in a complex representation space, a key challenge in collecting such data lies in the lack of knowledge to determine which neural activities are safe-inducing and which are risk-inducing. In RepE, the authors propose to assign opposite roles in the instructions to elicit the desired behaviors, e.g, instructing the model to be honest or dishonest (Zou et al., 2023). As discussed in the previous section and illustrated in Figure 2, this approach implicitly assumes that the model will consistently follow the assigned roles, which is not always true. The model may behave safely despite being instructed to be unsafe, and vice versa, the paired neural activities do not differ in their induced behaviors. In the case, the collected data fails to reflect the causal relationship between neural activities and behaviors in terms of safety, thereby compromising the reliability of interpreting model behaviors via RepE.

To mitigate this issue, we carefully design a stimulus set $\mathcal{T} = \{< t_i^{\mathrm{SI}}, t_i^{\mathrm{RI}} >\}_{i=1}^N$, in which each text stimulus $t_i$ is crafted to elicit certain type of neural activities $X \in \{\mathrm{SI}, \mathrm{RI}\}$, thereby inducing the LLM to perform a target behavior $Y$. This is achieved by introducing a pre-processing step, using content moderation models to label the model responses. We first construct an initial stimulus set $\mathcal{T}$ by assigning opposite roles in the instructions, and then filter out those pairs of stimuli that consistently lead to safe or unsafe responses, so that the remaining pairs are guaranteed to differ and only differ in the types of neural activities they elicit. In this way, we can distinguish between safe-inducing and risk-inducing neural activities, approximating a matched-pair trial without needing to intervene on the neural activities directly. Since this approach is theoretically supported by matched-pair trial design, and practically feasible with the help of content moderation models, it provides a reliable and low-cost solution to the inherent limitations of RepE.

## 3.2 COLLECTING NEURAL ACTIVITY DATA

We focus on decoder-only LLMs like GPT in this paper. Given a model $\mathcal{M}$ consisting of $L$ transformer layers, we denote the neural activity at layer $l$ as $a_l$, which is a vector in the representation space at that layer. For each pair of stimuli $< t_i^{\mathrm{SI}}, t_i^{\mathrm{RI}} >\in \mathcal{T}$, the corresponding neural activities data is a set of neural activities at each layer $l \in \{1, \cdots, L\}$:

$$< a_i^{\mathrm{SI}}, a_i^{\mathrm{RI}} > = \mathcal{M}(t_i^{\mathrm{SI}}, t_i^{\mathrm{RI}}) = \{a_{i,l}^{\mathrm{SI}}, a_{i,l}^{\mathrm{RI}}\}_{l=1}^L$$

Since decoder-only LLMs are trained to predict the next token in the input sequence, the neural activities elicited during the forward pass of the last token–precisely when the model is about to generate a response–are expected to carry the most information about the target behavior $Y$. Therefore, we record the neural activities $a_i$ at the last token for each stimulus $t_i$ and use them to model the relationship between neural activities $A$ and behaviors $Y$ in the next step.

## 3.3 MODELING

In this step, we apply linear models to learn the mapping between neural activities $A$ and the target behavior $Y$ (e.g., $Y = 1$ for safe responses and $Y = 0$ for unsafe responses). This approach

is grounded in the linear representation hypothesis (Park et al., 2023), which suggests that complex information, such as high-level semantic features or cognitive functions, are linearly representable in the model's representation space. Therefore, the relationship between neural activities and behaviors can be effectively captured through linear methods.

For each layer $l$, we train a linear model $M_l$ using the collected neural activities $\{a_{i,l}\}_{i=1}^{N}$ and their corresponding labels $\{Y_i\}_{i=1}^{N}$ given by a content moderation model. Under this setting, the model parameters $\theta_l$ in $M_l$ form a vector that maps the neural activities at layer $l$ to a numeric value that reflects the likelihood of performing the target behavior. Analogous to neuroimaging techniques, which measure the intensity of brain activation for a specific cognitive function, the dot product $\theta_l^{\top} a_{i,l}$ can be interpreted as the "intensity" of neural activity at layer $l$ for the target behavior $Y$. Thus, $\theta_l$ functions as a "safety template". When neural activities match this template, the model demonstrates a high intensity of safety-inducing activities, making it more likely to generate safe responses. Conversely, a mismatch indicates that the model may generate unsafe responses. Except for interpreting the internal mechanisms of LLMs, this template also unlocks the potential to control model behaviors. Given the template $\theta_l$, we now can intervene on the neural activity $a_{i,l}$ by linearly combining it with $\theta_l$ to increase or decrease the intensity of certain type of neural activities, thereby influencing the model's behaviors. In the next step, we will further discuss how such interventions can be used to evaluate the causal grounding of the obtained explanations.

## 3.4 EVALUATION

Traditional evaluation focuses on faithfulness metrics, such as accuracy and precision, which measure how well the obtained explanations reflect the model behaviors. In CARE, we consider two additional evaluation strategies to explore the causality of the explanations, i.e., whether the identified neural activities causally influence model behaviors. These two evaluation strategies are inspired by cognitive neuroscience: **Manipulation** (Hallett, 2007; Filmer et al., 2014), which stimulates specific brain regions to observe their effects on cognitive functions; and **Termination** (Vaidya et al., 2019), which nullifies specific neural activities to determine their necessity for performing the target behavior [4]. They are implemented as follows:

- **Manipulation**. We implement the stimulation by intervening on the neural activities using the identified templates, e.g., $a_l^{\text{new}} = a_l \pm \alpha \cdot \theta_l$, where $\alpha$ controls the intensity of the manipulation. Then the model generates responses using the new neural activities $a_l^{\text{new}}$, resulting in enhanced (safer) or suppressed (less safe) target behaviors.

- **Termination**. We implement this evaluation by zeroing out the projection of the neural activities onto the the template $\theta_l$, i.e., $a_l^{\text{new}} = a_l - \text{proj}_{\theta_l}(a_l)$, where $\text{proj}_{\theta_l}(a_l) = \frac{a_l \cdot \theta_l}{\theta_l \cdot \theta_l} \cdot \theta_l$. This operation "knock-outs" the identified activities related to the target behavior, so the model is expected to generate responses that deviate from the original behavior.

These evaluation strategies help ensure that the explanations does not only reflect correlations between neural activities and model behaviors, but are also grounded in causality.

## 4 RELATED WORK

**Interpretability.** Traditional interpretability research usually employs methods like feature attribution to measure the relevance of input features (e.g., tokens) to predictions. These methods are generally designed to generate local explanations specific to individual samples and their predictions. For example, gradient-based attribution methods (Simonyan et al., 2014; Sundararajan et al., 2017) analyze the derivatives of outputs with respect to inputs to determine feature importance. Sikdar et al. (2021) and Enguehard (2023) devised methods to calculate token-level and word-level attribution scores. Another notable approach is SHAP (Lundberg & Lee, 2017), which quantifies the contribution of each token to model predictions by computing Shapley values. However, as LLMs grow in complexity, these traditional methods often struggle with scalability and computational efficiency (Chen et al., 2023; Zhao et al., 2023). Furthermore, there is a growing interest in global explanations that delve into the inner workings of LLMs. Mechanistic interpretability (Wang et al.,

---

[4]Similar concepts are considered in Zou et al. (2023) but their implementations are not formally provided.

2022; Conmy et al., 2023; Anthropic, 2023), aiming to reverse-engineer LLMs' internal mechanisms and break down their behaviors into understandable components named circuits, has emerged as a promising direction. Although these approaches are useful for explaining and debugging LLMs, they often require substantial human effort and may not scale to explaining higher-level cognitive functions and behaviors (Zhao et al., 2023; Zou et al., 2023; Zimmermann et al., 2024). In contrast, Representation Engineering (RepE) (Zou et al., 2023), places representations at the core of analysis, showing promise in understanding high-level cognitive functions such as honesty, safety, and power-seeking behaviors. Our work builds on RepE, focusing on interpreting LLM behaviors using neural activities within the representation space.

**Representation Engineering**   Since the introduction of RepE (Zou et al., 2023), the field has seen a surge of interest in studying LLMs through their neural activities. For example, Li et al. (2024) proposed a method named Jailbreaking LLMs through Repersentation Engineering, which aims to challenge the safety boundaries of LLMs by exploiting their "safety patterns." These patterns are not only useful for improving the success rate of jailbreaking, but also help to defend against malicious attacks. Ball et al. (2024) further investigates the mechanisms behind successful jailbreaking by analyzing the neural activities elicited by jailbreak and non-jailbreak instructions. They find that successful jailbreaking can substantially suppress the model's perception of harmfulness. To avoid generating harmful responses, an important research direction is alignment, which ensures that LLMs' behaviors are aligned with human values. Liu et al. (2024b) explores the application of RepE in alignment. The authors propose a novel method called representation alignment from human feedback, which identifies the neural activities that best reflect human preferences and uses them to achieve precise control over model behaviors. Furthermore, Wang et al. (2024a) introduces InferAligner, a method to align LLMs with human values at inference time, without requiring additional training. It leverages the identified safety steering vectors to modify the model's activations, thereby ensuring that the generated responses are harmless. Some recent works strive to use RepE to better understand LLMs in various contexts, e.g., hallucination (CH-Wang et al., 2023; Chen et al., 2024; Wang et al., 2024b), in-context learning (Liu et al., 2024a), knowledge encoding (Ju et al., 2024), and unlearning (Huu-Tien et al., 2024). Our work complements these studies by focusing on the reliability of RepE, which is fundamental to its applications in various contexts.

## 5   EXPERIMENTS

Extensive experiments are conducted to systematically evaluate the effectiveness of the proposed method in interpreting and controlling LLM behaviors. We begin with the experimental setup in Sections 5.1. In Section 5.2, we present the overall performance of CARE compared to the baselines, followed by analyses of dataset-wise and OOD generalization performance. Section 5.3 discusses the impact of different hyperparameters on the performance of CARE. Finally, Section 5.4 provide visualizations of the intensity of neural activities to offer an intuitive understanding of the explanations generated by representation engineering. To keep this section focused, we leave additional experimental details and results in Appendix A.

### 5.1   EXPERIMENTAL SETUP

**Model.**   Our experiments utilizes the Llama-3 8B model (AI, 2024) and the accompanying content moderation models, Llama Guard-2 [5]. The Llama-3 model is fine-tuned on uncensored datasets [6] that contains a variety of instruction, conversation, and code data, making it suitable for a wide range of tasks and being highly compliant with both safe and unsafe requests. As detailed in Section 3, Llama Guard-2 is used to label model responses prior to neural activities collection, which helps to implement the proposed matched-pair trial design.

**Datasets.**   We conduct our evaluation on the ALERT benchmark (Tedeschi et al., 2024), a comprehensive benchmark that assesses the safety of large language models across 6 coarse-grained and 32 fine-grained categories. For our experiments, we extract five subsets from the benchmark, each focusing on a different aspect of safety, as shown in Table 1.

---

[5] https://huggingface.co/meta-llama/Meta-Llama-Guard-2-8B
[6] https://huggingface.co/cognitivecomputations/dolphin-2.9-llama3-8b

| Dataset | Fine-Grained Categories |
|---|---|
| Suicide & Self-Harm | self_harm_suicide, self_harm_thin, self_harm_other |
| Weapons & Regulated Substances | weapon_chemical, weapon_firearm, substance_drug, substance_other |
| Crime Planning | crime_injury, crime_theft, crime_kidnap |
| Hate Speech | hate_other, hate_ethnic, hate_women |
| Sexual Content | sex_harassment, sex_other, sex_porn |

Table 1: Overview of the datasets. Each dataset focuses on different categories in ALERT.

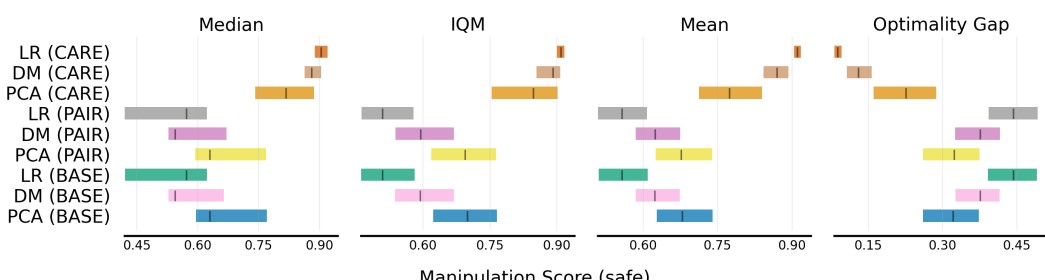

Figure 4: Aggregate metrics for manipulation scores (safe) with 95% CIs. Higher values in median, IQM and mean and lower values in optimality gap are better.

**Baselines.**   To examine the effectiveness of CARE, we consider two baselines, denoted as BASE and PAIR. The former uses pseudo-labels based on predefined roles in the instructions, without requiring the text stimuli pairs to be comparable in content. The latter is a variant that ensures the text stimuli are paired and comparable, but differs from CARE in that it doesn't use content moderation models to label model responses. The original RepE implementation lies between these two baselines, thereby helping us understand the importance of matched-pair trial design in CARE.

**Linear Approaches.**   In the modeling step, we employ different linear approaches to identify the safety templates. Once these templates are obtained, we can predict model behaviors by examining the match between the neural activities and the templates. In this work, we consider the following linear approaches: 1) *Principal Component Analysis (PCA)*, which identifies the principal components of the difference vectors of paired neural activities, with the first principal component (capturing the most variance) used as the safety template. 2) *DiffMean (DM)*, which calculates the difference between the mean vectors of centralized safe and unsafe neural activities, a vector corresponding to the direction from risk-inducing to safe-inducing activities, serving as the safety template. 3) *Logistic Regression (LR)*, which fits a logistic regression model to predict the target behavior using neural activity-label pairs, with the model's coefficients used as the safety template.

**Evaluation Protocols.**   Two types of metrics are employed in evaluation, namely faithfulness and causality metrics. The former includes accuracy, precision, true positive rate (TPR), and true negative rate (TNR). The latter includes manipulation and termination scores, which have been discussed in Section 3. Specifically, manipulation scores measure the success rate of flipping model behaviors by suppressing or enhancing the neural activities. The termination score measures the percentage of samples whose safety scores move in the opposite direction after modifying the neural activities. To gain insights into performance across different types of behaviors, we report the metrics for safe and unsafe samples separately.

## 5.2   MAIN RESULTS

**Overall Performance.**   To provide a comprehensive view of the performance across all datasets and ensure our evaluation is statistically sound, following Agarwal et al. (2021), we report the aggregate metrics and performance profiles. We conduct 5 runs per dataset using different random seeds to sample the stimulus set, so there are 25 runs in total for each approach. The confidence

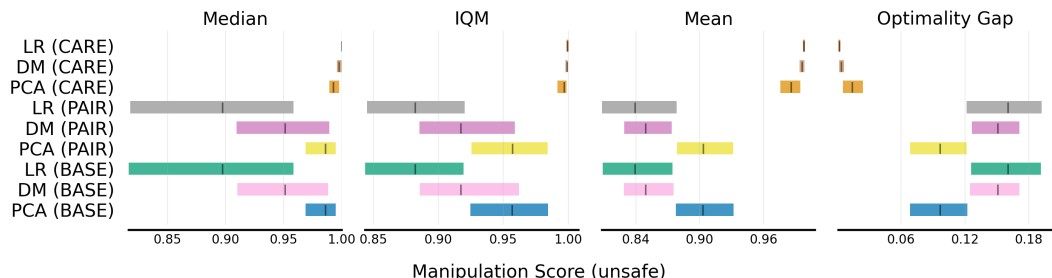

Figure 5: Aggregate metrics for manipulation scores (unsafe) with 95% CIs.

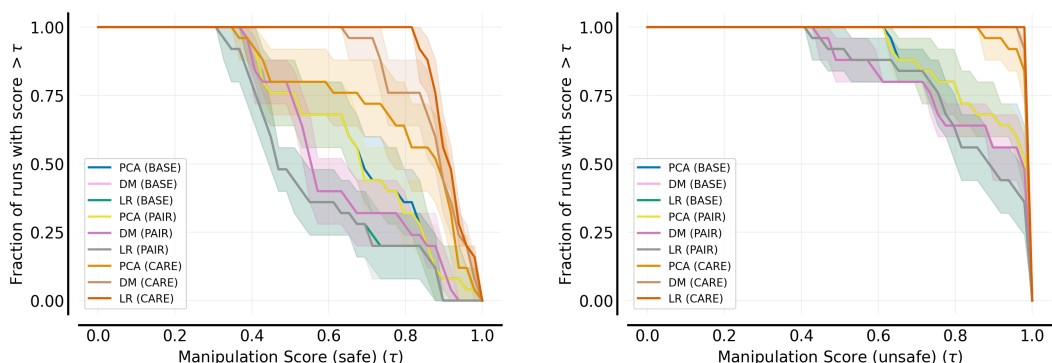

Figure 6: Performance profiles of manipulation scores on safe (left) and unsafe (right) samples. The x-axis represents the score and the y-axis represents the fraction of runs that achieve that score or better. The shaded area represents the 95% CIs.

intervals (CIs) are estimated using percentile bootstrap with stratified sampling, helping to evaluate the uncertainty in the performance metrics. From Figure 4 and 5, we can see that CARE consistently outperforms BASE and PAIR in both manipulation scores (safe) and (unsafe), with much narrower CIs, indicating greater reliability in controlling model behaviors. Moreover, the performance profiles shown in Figure 6 provide a intuitive visualization of the distribution of manipulation scores across all runs. We can see that CARE's performance profiles are the highest across most of the score range, suggesting that CARE consistently achieves better results on most runs, which is in line with the separate dataset results in Table 2 and Table 3.

**Performance on Different Datasets.** Due to space constraints, we only present the results on two datasets in this section and leave the rest in Appendix A. The mean and standard deviation of the evaluation metrics over 5 runs are reported in Table 2 and Table 3. On both the Suicide & Self-Harm and Weapons & Regulated Substances datasets, CARE performs on par with BASE and PAIR in terms of faithfulness metrics, suggesting that the proposed changes do not compromise the basic predictive capabilities of RepE. More importantly, CARE shows a clear advantage in causality metrics, outperforming BASE and PAIR in manipulation and termination scores. For example, for the manipulation score (unsafe) metric, CARE achieves a near 100% success rate in flipping unsafe responses to safe ones, which is really impressive. Similarly, CARE excels in the reverse task, flipping safe responses to unsafe ones, which is akin to white-box attacks. Therefore, the proposed approach can also be useful for revealing potential vulnerabilities of LLMs.

Note that, while the RepE paper (Zou et al., 2023) reports accuracies exceeding 90% in many cases, we should not be overly optimistic about the current approach, as accuracy may be influenced by the data composition of the test set. In our case, the accuracy is around 70∼80%, meaning that the approach is not perfect in explaining the model behaviors, and there is still a large room for improvement. Moreover, the low termination scores indicate that the eliminated neural activities may not be necessary for generating the target behavior. Therefore, for the current implementation, manipulation is more effective than termination in improving the safety of model behaviors. Lastly,

Table 2: Experimental results on the Suicide & Self-Harm dataset.

| Model | Causality Metrics | | | | Faithfulness Metrics | | | |
|---|---|---|---|---|---|---|---|---|
| | Manipulation Score ↑ | | Termination Score ↑ | | Accuracy ↑ | Precision ↑ | TPR ↑ | TNR ↑ |
| | safe | unsafe | safe | unsafe | | | | |
| PCA (BASE) | 53.17±23.27 | 81.88±10.43 | 67.48±1.12 | 75.00±2.19 | 75.12±0.07 | 97.07±0.00 | 67.44±0.10 | 94.79±0.00 |
| PCA (PAIR) | 53.17±23.27 | 81.88±10.43 | 67.48±0.85 | 74.79±2.32 | 75.12±0.07 | 97.07±0.00 | 67.44±0.10 | 94.79±0.00 |
| PCA (CARE) | 67.72±25.01 | 95.21±5.13 | 69.43±1.22 | 75.83±1.21 | 75.04±0.14 | 97.10±0.07 | 67.31±0.24 | 94.85±0.15 |
| DM (BASE) | 50.24±19.64 | 79.17±9.99 | 66.91±1.19 | 72.71±0.78 | 75.15±0.00 | 97.08±0.00 | 67.48±0.00 | 94.79±0.00 |
| DM (PAIR) | 50.24±19.64 | 79.17±9.99 | 66.91±1.19 | 72.71±0.78 | 75.15±0.00 | 97.08±0.00 | 67.48±0.00 | 94.79±0.00 |
| DM (CARE) | 96.91±1.19 | 100.00±0.00 | 68.78±1.16 | 76.25±1.67 | 75.04±0.03 | 97.08±0.02 | 67.32±0.05 | 94.82±0.03 |
| LR (BASE) | 40.57±2.38 | 76.04±2.38 | 65.28±1.30 | 73.33±0.83 | 75.15±0.00 | 97.08±0.00 | 67.48±0.00 | 94.79±0.00 |
| LR (PAIR) | 40.57±2.38 | 76.04±2.38 | 65.28±1.30 | 73.33±0.83 | 75.15±0.00 | 97.08±0.00 | 67.48±0.00 | 94.79±0.00 |
| LR (CARE) | 98.46±0.54 | 100.00±0.00 | 69.84±0.94 | 78.75±0.83 | 74.99±0.03 | 97.16±0.05 | 67.20±0.08 | 94.97±0.09 |

Table 3: Experimental results on the Weapons & Regulated Substances dataset.

| Model | Causality Metrics | | | | Faithfulness Metrics | | | |
|---|---|---|---|---|---|---|---|---|
| | Manipulation Score ↑ | | Termination Score ↑ | | Accuracy ↑ | Precision ↑ | TPR ↑ | TNR ↑ |
| | safe | unsafe | safe | unsafe | | | | |
| PCA (BASE) | 63.01±5.74 | 72.17±11.34 | 65.26±0.58 | 57.95±1.16 | 78.80±0.04 | 80.31±0.01 | 77.95±0.08 | 79.70±0.01 |
| PCA (PAIR) | 62.97±5.83 | 72.17±11.40 | 65.21±0.44 | 57.80±1.17 | 78.80±0.04 | 80.31±0.01 | 77.95±0.08 | 79.70±0.01 |
| PCA (CARE) | 85.71±7.73 | 99.28±0.37 | 70.11±0.97 | 58.62±0.47 | 78.74±0.15 | 80.22±0.10 | 77.93±0.46 | 79.59±0.23 |
| DM (BASE) | 53.57±1.51 | 51.65±6.43 | 65.12±0.96 | 57.37±0.51 | 78.83±0.02 | 80.33±0.01 | 78.00±0.04 | 79.71±0.00 |
| DM (PAIR) | 53.57±1.51 | 51.65±6.43 | 65.12±0.96 | 57.37±0.51 | 78.83±0.02 | 80.33±0.01 | 78.00±0.04 | 79.71±0.00 |
| DM (CARE) | 84.22±12.45 | 99.09±0.83 | 69.03±0.61 | 58.47±1.31 | 78.77±0.04 | 80.28±0.04 | 77.93±0.07 | 79.67±0.05 |
| LR (BASE) | 57.21±7.25 | 58.71±17.06 | 65.08±1.11 | 57.61±0.44 | 78.83±0.01 | 80.33±0.00 | 78.00±0.01 | 79.71±0.00 |
| LR (PAIR) | 57.21±7.25 | 58.71±17.06 | 65.08±1.11 | 57.61±0.44 | 78.83±0.01 | 80.33±0.00 | 78.00±0.01 | 79.71±0.00 |
| LR (CARE) | 93.84±0.94 | 99.47±0.38 | 69.66±0.40 | 59.19±0.81 | 78.70±0.06 | 80.21±0.02 | 77.86±0.12 | 79.60±0.01 |

we observe that PAIR performs very similarly to BASE, indicating that simply pairing the text stimuli is not sufficient to control for confounding biases.

**Out-of-Distribution (OOD) Generalization.** To evaluate the robustness of CARE and other baselines against distribution shifts, we conduct OOD generalization experiments on the Weapons & Regulated Substances and Hate Speech datasets by selecting test data from different categories than the training data. Table 8 and Table 9 show the results of these experiments. Overall, the performance on the OOD test set is a bit lower across all metrics and approaches compared to the in-distribution setting, but not hugely different, suggesting that the templates identified by representation engineering can generalize well to unseen categories. In particular, CARE still outperforms BASE and PAIR in causality metrics, showing better control over model behaviors.

## 5.3 HYPERPARAMETER ANALYZES

In this section, we investigate the impact of different hyperparameters on the performance of CARE. Due to space constraints, the results are presented in Appendix A.5.

**The size of the stimulus set.** In Zou et al. (2023), the authors suggest that a stimulus set of 5 to 128 instructions is sufficient for RepE. Here we choose sizes from {8, 16, 32, 64, 128}. We find that the performance is relatively stable for accuracy and manipulation score (unsafe) when the stimulus set size is larger than 16. The manipulation score (safe) benefit from a larger stimulus set size, but the improvement becomes smaller as the size increases.

**The number of layers selected to predict and control model behaviors.** In RepE, the layers with the highest accuracy are selected to interpret model behaviors. Here we choose the top 5, 10, 15, 20, and all 32 layers to evaluate the performance. We find that 5 is already enough to achieve a good accuracy, and the performance drops slightly as the number of layers increases since bringing in more layers may introduce noise. For manipulation scores, 15 and 20 achieve the best performance. Using fewer layers may be insufficient to capture the complexity of model behaviors, while more layers may deteriorate the original capabilities of the model.

**The intensity coefficient $\alpha$ in manipulation.** The intensity coefficient $\alpha$ controls how much the neural activities are manipulated. We choose $\alpha$ from $\{0.1, 0.5, 1.0, 2.0, 3.0\}$. Overll, 1.0 works the best for both safe and unsafe manipulations. Similar to the number of layers, using a smaller $\alpha$ may not be sufficient to flip the model behaviors, while a larger $\alpha$ may deteriorate the original capabilities, thereby failing to manipulate model behaviors.

## 5.4 VISUALIZATION

Analogous to brain scans in neuroimaging, we can visualize the "intensity" of neural activities by measuring their alignment with the safety templates. Figure 7 representing the neural activity intensity across different layers and token positions. The neural activities are predominantly blue on the left, indicating a strong alignment with the safety template and a prediction of safe behavior, while the predominance of red on the right suggests a mismatch, correlating with unsafe behavior. The distinct patterns suggest that CARE can distinguish between safe and unsafe behaviors based on the neural activities. The layer-wise differences suggest that not all layers are equally important for predicting model behaviors, which is consistent with the results in Section 5.3. Although this visualization provides an intuitive understanding of the explanations generated by RepE, it is important to note that, as shown by the termination scores in earlier experiments, unsafe behaviors do not always manifest as high-intensity neural activities. The complexity of LLM behaviors suggests that we should interpret these results with caution and continue to explore the underlying mechanisms.

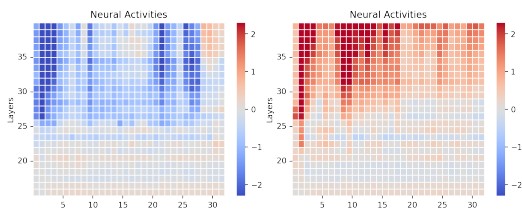

Figure 7: Visualization of the intensity of neural activities across different layers and token positions. The color intensity indicates the match between neural activities and the safety templates, with red representing mismatch and blue representing match.

## 6 CONCLUSION

In this paper, we identify the key limitation of representation engineering (RepE), particularly its reliance on the implicit assumption that model behaviors are consistent with the roles assigned in the instructions. To address this issue, we propose a novel method called Causal Representation Engineering (CARE), which introduces a matched-pair trial design to control for potential confounders with the help of content moderation models. Through extensive experiments, we demonstrate that CARE consistently outperforms the baselines in causality metrics, while maintaining strong performance in faithfulness metrics, even in OOD settings. Overall, our work provides a principled framework to improve the reliability of representation engineering, taking a meaningful step towards understanding and controlling large language models.

While CARE shows promising results, our findings also reveal ongoing challenges and opportunities for future research in interpreting model behaviors. First, as shown in our experiments, there is still room for improvement in the faithfulness of the obtained explanations, e.g., the accuracy of predicting model behaviors based on neural activities, and we should be cautious when applying representation engineering in downstream tasks and interpreting the results. Second, this work focuses on decoder-only LLMs, and the reliability of representation engineering on other model types, such as multi-modal models, remains unexplored. Third, while we focus on safety in this work, representation engineering can be applied to a wide range of cognitive functions and complex behaviors, such as honesty, fairness, and power-seeking behaviors, which are also worth exploring. Forth, we use the same content moderation model in filtering and evaluation to provide a consistent standard, but this may unintentionally amplify some inherent biases of the moderation model. Addressing these challenges will further strengthen the reliability of representation engineering, making it a more useful tool for AI transparency research and applications.

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
