# OpenReview forum: "Rethinking The Reliability of Representation Engineering in Large Language Models"
_ICLR.cc/2025/Conference — Submitted to ICLR 2025_

### Official Review · Reviewer_frYB · 2024-11-01

**Soundness:** 3
**Presentation:** 4
**Contribution:** 1
**Rating:** 5
**Confidence:** 5

**Summary:**

The paper follows up on work on representation engineering (RepE) which shows that activations of language models can be used to interpret and steer their behaviours. The main contribution is filtering the activations collected to find the intervention vector in the LAT methodology proposed in the original RepE paper, removing instances where the model does not follow the instructions provided using Llamaguard. The experiments are principled, focusing on causal evaluations of manipulation and termination in the RepE paper, using the ALERT dataset for studying diverse behaviours rigorously, and showing the effect of varying crucial design choices like the modelling method, number of samples used, number of layers, and the intervention intensity coefficient.

**Strengths:**

1. This work does a great job of rigorously analysing the design choices of the LAT methodology proposed in the RepE paper, which has higher importance because this is done poorly in the original RepE paper. This can be useful for practitioners seeking to apply RepE.

2. The paper is well written and easy to follow, barring some concerns regarding how the contributions are positioned mentioned below.

**Weaknesses:**

1. The main contribution of this work is limited, with its impact decreasing over time. The paper shows the importance of filtering instances where the model does not follow instructions before finding the LAT vector. The importance of this contribution is diluted as models get better at instruction following, as lesser samples would be filtered.

2. The contribution can seem misleading based on the writeup of the paper, as it does not properly contextualise the new contributions on top of the original RepE paper. For example:

a) The abstract/intro make it sound like matched-pair trial design is a contribution of this work. However, the RepE paper already uses this setup without making the connection to the naming convention used in control trial design. The only contribution seems to be filtering out the cases where the model fails to follow the instruction.

b) Manipulation and Termination are already studied as evaluations in RepE (see Figure 12 in Section 5). Lines 98-100 and Section 3.4 (especial L240-242) make it sound like these are contributions of this work, which is not true.

c) It is unclear how BASE (”using pseudo-labels based on predefined roles in the instructions, without requiring the text stimuli pairs to be comparable in content”) accurately reflects the methodology in RepE. To the best of my knowledge, RepE also uses contrastive minimal pair prompts to collect the stimuli, differing only in the behaviour to ablate.

Usually one would do this in the related work section. However, L280 in related work section just says:  "Our work builds on RepE, focusing on interpreting LLM behaviors using neural activities within the representation space", but this is exactly what RepE does so its not clear what this work builds on top. The only marginal contribution I can see is labelling model responses using llama-guard and filtering to the ones that follow the desired instructions before performing RepE, which is weak as pointed in W1.

3) The headline figure (which is Figure 2, as Figure 1 doesn't convey much information) is hard to interpret. It is unclear what the task is in figure 2 i.e. what is RepE being used for, which dataset, how many classes is the classification over etc. It is also unclear what the “RepE” method refers to. RepE is the paper whereas LAT is the method proposed in the paper.  Finally, at this point in the paper, it is unclear what is ‘inconsistent behaviour’ it would be useful to provide a concrete example in the intro, before figure 2.

**Questions:**

1. What percentage of prompts do different LLMs not follow the instructions for? How does this vary based on the model's instruction following capabilities, and scaling?

2. Have you considered submitting this work to the ICLR Blogpost track or TMLR? I think these venues are more suitable for work like this with rigorous analysis and reproduction, but limited technical novelty or new ideas.

3. Line 133-134 is unclear. why is it infeasible to intervene on neural activities? How is intervene defined here can you be more specific? The original RepE paper does one form of intervention, by adding the vector found using LAT to the representation, why does this not count?

4. In section 3.3 on modelling, the linear model setup described is supervised, whereas RepE also focuses on unsupervised methods. It would be nice to make it explicit why this design choice was made.

5 (typo/minor). Section 5 L308-310 references are broken.


**Update after discussion phase**: I thank the authors for the detailed discussion. After extensive followups, the authors confirmed that the only 'confounding factor' being controlled for here is the model not always following instructions. This is the only methodological contribution of this paper on top of the original Representation Engineering paper, and the rest is causal and neuroscience motivation. Thus I stick with my original review and score.

---

> ### Author Response · Authors · 2024-11-24
> **Response to Reviewer frYB**
>
> Thank you for your detailed feedback. We appreciate your comments, which have helped us better clarify our contributions. Below, we address each of your concerns in detail.
>
> > The impact might be weakened as LLMs scale up and improve their instruction-following capabilities.
>
> We appreciate your thoughtful observation. While instruction-following capabilities may improve with scale, but this improvement does not automatically translate to enhanced reliability in representation engineering. Ignoring the inherent limitation of being confounded by spurious correlations could lead to overconfidence in this approach.
>
> To address your concern, we conducted additional experiments on 1.1B and 13B models, comparing them to our original 8B results:
>
> | Model       | Manipulation Score (safe) | Manipulation Score (unsafe) | Termination Score (safe) | Termination Score (unsafe) | Accuracy     | Precision    | TPR          | TNR          |
> | ----------- | ------------------------- | --------------------------- | ------------------------ | -------------------------- | ------------ | ------------ | ------------ | ------------ |
> | 1.1B (PAIR) | 30.90 ± 25.32             | 96.45 ± 4.03                | 59.23 ± 1.02             | 64.30 ± 2.16               | 46.42 ± 1.05 | 67.86 ± 1.02 | 40.34 ± 1.57 | 59.65 ± 0.91 |
> | 1.1B (CARE) | 72.88 ± 0.17              | 82.43 ± 13.08               | 60.69 ± 1.14             | 71.21 ± 2.80               | 50.74 ± 0.46 | 71.04 ± 0.98 | 47.71 ± 0.78 | 57.33 ± 2.37 |
> | 8B (PAIR)   | 40.57 ± 2.38              | 76.04 ± 2.38                | 65.28 ± 1.30             | 73.33 ± 0.83               | 75.15 ± 0.00 | 97.08 ± 0.00 | 67.48 ± 0.00 | 94.79 ± 0.00 |
> | 8B (CARE)   | 98.46 ± 0.54              | 100.00 ± 0.00               | 69.84 ± 0.94             | 78.75 ± 0.83               | 74.99 ± 0.03 | 97.16 ± 0.05 | 67.20 ± 0.08 | 94.97 ± 0.09 |
> | 13B (PAIR)  | 18.81 ± 1.39              | 45.56 ± 1.99                | 84.60 ± 1.08             | 82.67 ± 0.54               | 65.36 ± 0.05 | 88.92 ± 0.06 | 60.53 ± 0.08 | 78.87 ± 0.14 |
> | 13B (CARE)  | 21.75 ± 0.81              | 49.33 ± 3.03                | 85.24 ± 0.16             | 82.00 ± 0.44               | 65.45 ± 0.05 | 88.81 ± 0.07 | 60.77 ± 0.14 | 78.56 ± 0.21 |
>
> While CARE consistently outperforms PAIR, the performance gap narrows with model scale. Interestingly, manipulation scores is lower for the 13B model, not higher, suggesting that larger models may be more resistant to manipulation. Therefore, it is important to study the reliability of representation engineering as models scale up.
>
> The impact of our work is not weakened. The new results highlight two exiting and promising directions:
> - **For smaller models**: CARE provides a principled and low-cost approach to enhance reliability, making them suitable for safety-sensitive applications with constrained resources.
> - **For larger models**, where manipulation remains challenging, future work could explore techniques in emerging research areas such as red-teaming to further improve safety.
>
> We have added a discussion of these results and their implications in Appendix A.7. If you have further references or suggestions, we would be glad to include and discuss them. Thank you again for your valuable insight.
>
> > The RepE paper already uses this setup without making the connection to the naming convention used in control trial design.
>
> We respectfully disagree with this opinion. The RepE paper does not  implement matched-pair trial design and is unclear in its setting. For example,
> - In the Honesty experiment of the RepE paper (Sec. 4.3), paired stimuli is used but do not consistently elicit opposing behaviors.
> - In the Harmfulness experiment of the RepE paper (Sec. 6.2), the stimuli are not guaranteed to be paired at all.
>
> This ambiguity places RepE’s approach in a gray area between our "BASE" and "PAIR" baselines. Importantly, this lack of methodological rigor and understanding of causality is a key reason for the unreliability of representation engineering.
>
> Our work addresses this ambiguity by introducing a rigorous definition and implementation of matched-pair trial design, which is essential for isolating causal effects. This methodological clarity and precision enhance the reproducibility and reliability of representation engineering.

---

> ### Author Response · Authors · 2024-11-24
> **Response to Reviewer frYB (cont.)**
>
> > Manipulation and Termination are already studied as evaluations in RepE.
>
> We did not list and will not claim manipulation and termination as our contributions. Instead, our goal was to formalize these metrics and distinguish them from traditional evaluation metrics (e.g., accuracy, precision).
>
> The RepE paper references these concepts but does not provide formal definitions, and no corresponding implementations can be found in its open-source code. Explicitly formulating and explaining these metrics serve to enhance the clarity of the evaluation and help readers better understand our experimental results.
>
> To avoid any misunderstanding about the contributions, we have carefully added a statement in the revised manuscript emphasizing that the RepE paper has considered similar concepts.
>
> > L280 in related work section just says: "Our work builds on RepE, focusing on interpreting LLM behaviors using neural activities within the representation space", but this is exactly what RepE does.
>
> This sentence might appear unclear when taken out of context, and we appreciate the opportunity to clarify its intent.
>
> This statement appears at the end of the first paragraph in the related work section, which introduces various interpretability techniques for LLMs. It serves to situate our work within the representation engineering paradigm, highlighting that our research builds on this specific interpretability technique.
>
> In the subsequent paragraph, we delve deeper into recent advancements in representation engineering, which primarily focus on applying this method to different scenarios (e.g., alignment, jailbreaking, hallucinations). We explicitly distinguish our work by emphasizing its focus on **the reliability aspect of representation engineering itself**, a novel and orthogonal research direction that has not been explored in prior work and is fundamental to its applications.
>
> We understand that the reviewer’s concern may stem from the ambiguity in the RepE paper. To address this, we have strengthened the comparison in the revised manuscript to ensure our focus on reliability is evident. Thanks for the feedback!
>
> >  It would be useful to provide a concrete example for "inconsistent behaviour" in the intro, before figure 2.
>
> This is a great suggestion! In the revised manuscript, we have included a concrete example (Figure 1) illustrating inconsistent behavior. This will better contextualize the vulnerabilities of representation engineering methods and make Figure 2 more accessible to readers.
>
> > Percentage of prompts do LLM fails to follow instructions.
>
> Approximately 50% of prompts exhibit instruction-following failures in our experiment, depending on the dataset being used. This includes cases where the model refused to follow an unsafe instruction, as well as cases where the model behaves unsafely even though it was asked to behave safely.
>
> > Why is it infeasible to intervene on neural activities?
>
> Intervening on neural activities is challenging due to their high dimensionality and lack of interpretability. We cannot directly modify individual dimensions in a way that reliably translates to desired changes in model behaviors. Representation engineering achieves intervention by learning a manipulation vector, but it does not reliably translate to desired changes in model behaviors either, as shown by the abundant experimental results.
>
> > The linear model setup described is supervised, whereas RepE also focuses on unsupervised methods.
>
> While the RepE paper emphasizes unsupervised methods (e.g., PCA), **its actual implementation is supervised**, as it requires labels to construct contrastive pairs for PCA.
>
> Interestingly, while the RepE paper observes superior performance with PCA, our experiments show that LR performs comparably or even better when confounding factors are effectively addressed. This finding provides new insights.
>
> > L308-310 references are broken.
>
> Thank you for catching this issue. We have corrected the references in the revised manuscript.

---

> > ### Comment · Reviewer_frYB · 2024-11-25
> >
> > Thank you for the detailed rebuttal.
> >
> > My main concerns remain. The main new contribution of this work still seems to be just filtering responses where the model does not follow the instructions when using Representation Engineering.
> >
> > 1. The original RepE paper might not be using the matched pair design on some applications (eg harmfulness as mentioned in the response) where relevant data wasn't available, but clearly does it in others. While it's useful to take into account that sometimes the target model might not follow contrasting instructions, this isn't a fundamental change in design.
> >
> > 2. Thank you for the new table. It seems to show that for larger models, the effect of the proposed change (filtering) decreases as expected. While I agree this can be an important design choice when trying to perform RepE on smaller models, I do think models are getting better at instruction following over time across scales, which dilutes the importance of this contribution.
> >
> > I do commend the extensive empirical analysis done to show the effect of this change, but do not think the contribution in itself is enough for acceptance to ICLR due to the aforementioned reasons. The paper does not really "rethink" the reliability of RepE as claimed, and instead adds a minor improvement on top by filtering responses that don't follow instructions. I think this misleads the reader about the contributions, and encourage the authors to be more specific in future versions.

---

> > > ### Author Response · Authors · 2024-11-27
> > > **Response to Reviewer frYB**
> > >
> > > Thank you very much for your further feedback. We appreciate the opportunity to address your concerns.
> > >
> > > > The original RepE paper might not be using the matched pair design ... but clearly does it in others. While it's useful to ..., this isn't a fundamental change in design.
> > >
> > > **We respectfully disagree with this characterization.** In our paper, we explicitly define the matched-pair design and explain why the original RepE paper did not implement it. If you believe the RepE paper indeed does matched-pair design, we would appreciate it if you could point to the specific sections or experiments that demonstrate this. To our knowledge, the original implementation lies in a gray area between our "BASE" and "PAIR" baselines.
> > >
> > > Moreover, the simplicity of a modification does not detract from its significance, particularly when it identifies and addresses a fundamental issue: the assumption of consistent behavior during neural activity collection. This assumption, if left unaddressed, undermines the reliability of representation engineering. Our extensive experimental results support this modification is not only theoretically sound but also practically effective. As such, it is not merely incremental but a foundational change to the methodology.
> > >
> > > > I do think models are getting better at instruction following over time across scales, which dilutes the importance of this contribution.
> > >
> > > We appreciate this perspective but interpret the results differently. As detailed in the previous reply, CARE consistently improves performance across all model sizes. While the margin of improvement decreases for larger models, this does not negate the importance of the contribution. Instead, it highlights an intriguing direction for future work: understanding and addressing the evolving challenges of representation engineering as models scale.
> > >
> > > Additionally, concerns about diminishing performance due to improved instruction-following capabilities apply equally to the original RepE methodology. This further underscores the importance of systematically investigating the reliability issues, as CARE aims to do.
> > >
> > > > The paper does not really "rethink" the reliability of RepE as claimed.
> > >
> > > We would like to clarify our intent behind the term “rethink.” As you noted, “This work does a great job of rigorously analysing..., which has higher importance because this is done poorly in the original RepE paper.” Our goal is precisely to highlight and address the ambiguities and limitations in the original RepE methodology. By identifying the sources of unreliability and providing a systematic framework to address them, we aim to refocus attention on the reliability of representation engineering—a critical but underexplored aspect. We believe this perspective invites the community to critically evaluate and improve RepE, which is beneficial to the research community.

---

> ### Comment · Reviewer_frYB · 2024-11-28
>
> > If you believe the RepE paper indeed does matched-pair design, we would appreciate it if you could point to the specific sections or experiments that demonstrate this
>
> In the RepE paper [1], see algorithm 1 line 7 and 8 where the experimental template and control template give rise to contrast vectors (line 10) that are used for steering, which is essentially the same as the matched pair design. See Section 4.3 as an application of this methodology to honesty. This discussion makes me wonder if the paper accurately reproduces the LAT method in RepE in their results at all.
>
> > the simplicity of a modification does not detract from its significance
>
> I agree with this statement. However, I earlier raised concerns regarding the significance of this contribution, hypothesising that the gain achieved by the proposed methodology change of filtering stimuli would diminish as instruction following capabilities of models improve. This has held true from the followup experiments provided, as quoted by the authors:
>
> > the margin of improvement decreases for larger models
>
> I am still willing to change my mind if you show a plot between instruction following capabilities on the X axis, and the gain from your methodology over contrast vectors / LAT in RepE on the Y axis, and if the slope is not negative after adding sufficient models with varying instruction following capabilities.
>
> > a foundational change to the methodology
>
> I strongly disagree with this characterisation and think this amounts to over claiming, which is one of the main issues with the paper.
>
> >  diminishing performance due to improved instruction-following capabilities apply equally to the original RepE methodology
>
> My concern was the relative diminishing gain over RepE with the proposed methodological change in this paper, and not diminished absolute results. Also, it is not clear from the results reported how the original RepE methodology leads to diminishing performance with instruction following to me. I dont think this claim can be made from the results shown in this paper.
>
> [1] Zou, Andy, et al. "Representation engineering: A top-down approach to ai transparency." arXiv preprint arXiv:2310.01405 (2023).

---

> > ### Author Response · Authors · 2024-11-28
> > **Response to Reviewer frYB**
> >
> > > In the RepE paper [1], see algorithm 1 line 7 and 8 ...
> >
> > Thank you for pointing this out. We now understand that there is a **misunderstanding** regarding the concept of "matched-pair design." What you refer to as "essentially the same as the matched-pair design" aligns more closely with **PAIR** as implemented in our baselines. In contrast, the matched-pair trial requires carefully controlling for confounding factors, which RepE clearly lacks. Please read the second and third paragraphs of Section 3.1 in our paper for reference.
> >
> > > The gain achieved by the proposed methodology change of filtering stimuli would diminish as instruction-following capabilities of models improve.
> >
> > Regarding the results on the 13B model, we interpret them differently. As a first step toward exploring the reliability of representation engineering, our results on the 8B model provide strong evidence for two key points: (1) the original implementation of RepE is not reliable, and (2) the proposed modification brings robust improvements. While the gain is smaller for the 13B model, we view this as an exciting direction for future work—how to improve representation engineering reliability as models scale and become more capable of instruction-following.
> >
> > > I am still willing to change my mind if you show a plot between instruction-following capabilities on the X-axis and the gain from your methodology ...
> >
> > We greatly appreciate your suggestion. Although we did not claim contributions on this specific aspect, we are willing to conduct additional experiments to address your concern. If time permits, we will include these results and share them with you for further discussion.
> >
> > > I strongly disagree with this characterisation and think this amounts to over claiming, which is one of the main issues with the paper.
> >
> > We respectfully disagree that overclaiming is a main issue in the paper. If you refer to our stated contributions on Pages 2–3, you will find that they are carefully phrased, evidence-based, and presented with appropriate caution. Furthermore, in the conclusion on Page 10, we explicitly acknowledge the limitations of the proposed approach. We agree that whether this constitutes a "foundational change" is subjective, and that's why we avoided using such language in the paper itself.
> >
> > We truly appreciate the opportunity to clarify these points and address any misunderstandings. Thank you for taking the time to provide feedback.

---

### Official Review · Reviewer_G4PV · 2024-11-02

**Soundness:** 2
**Presentation:** 3
**Contribution:** 3
**Rating:** 6
**Confidence:** 3

**Summary:**

This paper introduces CARE, a framework aimed at improving the reliability of representation engineering in large language models. The authors identify a key limitation in current RepE methods - the implicit assumption that models consistently follow assigned roles during neural activity collection. To address this, they propose using matched-pair trial design with content moderation models to control for confounding factors.

**Strengths:**

Strengths:

- The paper identifies a genuine limitation in RepE and proposes a creative solution using matched-pair trials and content moderation.
-  Demonstrates clear improvements over baselines, particularly in manipulation scores (98.46% for safe manipulations).
-  Tests the method across multiple tasks and provides detailed ablation studies.
- The work has immediate applications for improving AI safety and transparency.

**Weaknesses:**

The paper's fundamental theoretical weakness lies in its assumption about matched-pair trials in neural networks in my opinion. The authors propose matching inputs (text prompts) to identify and control for confounding factors. However, they fail to address several critical theoretical issues:

- a) Neural Network Non-linearity:

No insight is provided that similar inputs lead to meaningfully "matched" representations in deep networks. The non-linear transformations through multiple transformer layers could amplify small input differences. What are the bounds on how representation differences scale with input differences across network depth

- b) Causal Identification/claims:

The paper lacks formal criteria for when their matching procedure successfully isolates causal effects and if found it should provide discussion of potential hidden confounders that could violate the matching assumptions. Missing analysis of how representation distributions shift under their interventions.


- Methodology:


a) Evaluation Circularity:

Using similar types of models for filtering and evaluation could amplify shared biases. No discussion of how to validate results independent of content moderation assumptions.

b) Manipulation vs Termination:

Large unexplained gap between manipulation success (98.46%) and termination scores (~58%). This suggests either redundant causal pathways or methodological issues. Can you elaborate on this?

**Questions:**

- Can you provide some theoretical guarantees/insights for when input matching implies representation matching?

- Do you have a sense of what might explain the significant gap between manipulation scores (98%) and termination scores (58%)? Does this suggest redundant causal pathways?

- How sensitive are your results to the choice of content moderation model? Did you perform any ablation studies.

---

> ### Author Response · Authors · 2024-11-24
> **Response to Reviewer G4PV**
>
> Thank you very much for reviewing our paper and providing constructive feedback. We address each of your concerns below:
> > Causal Identification/claims.
>
> In CARE, our approach is based on a matched-pair trial design, which approximates a controlled experiment by ensuring that pairs of cases differ only in the type of neural activities (i.e., safe-inducing versus risk-inducing) while all other factors are held constant. By comparing the outcomes between these matched pairs, we can attribute the observed differences in model behaviors $Y$ to the neural activity $X$. This follows the standard definition of causality without making additional assumptions.
>
> More complex causal structures, e.g., those involving multiple confounding factors or mediators, might need sophisticated identification techniques or discussion on causal bounds as seen in causal inference literature. However, this is out of the scope of this paper.
>
> As for your concern on representation distribution shifts under interventions, we would appreciate any clarification on what specific aspects you are referring to. If you have relevant references or literature, we would be more than willing to incorporate and discuss these points in the revised version.
>
> > Theoretical guarantees/insights for when input matching implies representation matching in deep networks?
>
> Thank you for raising this thought-provoking point. We understand the concern that, due to the inherent complexity of deep networks, even with content moderation models filtering the stimuli, there may still be unobserved confounding factors that are not explicitly controlled and may introduce subtle challenges in ensuring the matching procedure.
>
> The goal of CARE is to provide a practical and approximate solution to isolate the causal effects of neural activity on model behaviors. By combining matched-pair trial design with content moderation models, we aim to minimize confounding effects to the extent possible within the constraints of the available tools and datasets. While this approach does not guarantee theoretical perfection, it offers a feasible framework to systematically evaluate and improve the reliability of representation engineering.
>
> We believe this broader question of ensuring perfect matching is a significant and ongoing challenge in causality research, one that extends beyond the scope of this paper. If there are existing studies or techniques addressing this specific issue, we would be eager to reference and discuss them in our revised manuscript. We also welcome further suggestions on how to address this limitation in future work.
>
> > Evaluation Circularity.
>
> Thank you for raising the concern about evaluation circularity and shared biases when using similar models for filtering and evaluation. This is indeed an interesting perspective, though we believe it is orthogonal to the core focus of this paper. The use of content moderation models (e.g., Llama-Guard) serves as a consistent standard to filter responses and reveal the limitations of representation engineering (RepE). Our objective was to demonstrate the vulnerabilities in RepE methods under a unified setup, not to develop a bias-free approach for evaluating LLMs.
>
> We agree that there is ongoing work, such as using LLMs as a judge, which raises similar concerns about biases in evaluation. This is indeed an interesting direction for future research, and we have revised the manuscript to highlight it in the conclusion.
>
> > Gap between the manipulation score and termination score.
>
> These two scores indeed serve different purposes and are not directly comparable in terms of a simple “higher is better” metric. Manipulation measures the ability to change behaviors by enhancing or suppressing neural activities, and termination reflects the necessity of those activities for generating specific behaviors.
>
> To clarify with an analogy: consider that smoking is neither a sufficient nor a necessary cause for developing lung cancer—some smokers never develop cancer, and some cancer patients have never smoked. However, smoking still increases the risk. Similarly, in our study, the identified neural activities are not always necessary for specific behaviors (as indicated by a low termination score), but they are highly effective for reversing model behaviors (as shown by a high manipulation score). These results suggest that manipulation may be a more suitable way for controlling model behaviors.

---

> > ### Author Response · Authors · 2024-11-24
> > **Response to Reviewer G4PV (cont.)**
> >
> > > Sensitivity to the choice of content moderation model.
> >
> > To address your concern, we conducted additional experiments on the Suicide and Self-Harm dataset using Google’s open-source ShieldGemma-9B model. The moderation model was used in two different steps: (1) to select the stimulus set and (2) to evaluate the results. To analyze the effects comprehensively, we constructed four experimental scenarios:
> > - LLaMA2LLaMA: LLaMA Guard-8B for both steps.
> > - Gemma2Gemma: ShieldGemma-9B for both steps.
> > - Gemma2LLaMA: ShieldGemma-9B for stimulus selection and LLaMA Guard-8B for evaluation.
> > - LLaMA2Gemma: LLaMA Guard-8B for stimulus selection and ShieldGemma-9B for evaluation.
> >
> > The results are shown below:
> >
> > | Model | LLaMA2LLaMA (safe) | LLaMA2LLaMA (unsafe) | Gemma2Gemma (safe) | Gemma2Gemma (unsafe) | Gemma2LLaMA (safe) | Gemma2LLaMA (unsafe) | LLaMA2Gemma (safe) | LLaMA2Gemma (unsafe) |
> > |-------|---------------------|----------------------|--------------------|-----------------------|--------------------|-----------------------|--------------------|-----------------------|
> > | PAIR  | 40.57 ± 2.38       | 76.04 ± 2.38        | 25.25 ± 2.25      | 73.49 ± 2.31         | 41.62 ± 2.00      | 78.80 ± 3.78         | 30.65 ± 2.16      | 72.92 ± 2.28         |
> > | CARE  | 98.46 ± 0.54       | 100.00 ± 0.00       | 54.75 ± 14.04     | 98.31 ± 2.81         | 78.15 ± 19.83     | 97.59 ± 2.02         | 74.31 ± 1.67      | 100.00 ± 0.00        |
> >
> > These results show that across all scenarios, the key findings remain consistent with the original setup (LLaMA Guard-8b for both steps), indicating that CARE is robust to the choice of content moderation models, further strengthening its practical applicability. We have updated the paper to reflect these new results in Appendix A.6.

---

> > > ### Comment · Reviewer_G4PV · 2024-11-25
> > > **Response to Authors**
> > >
> > > Thank you for taking the time to respond to my comments and questions. Most of my concerns have been clarified. There appears to be a strong consensus among the reviewers, and I am increasing my score from 5 to 6 to acknowledge the authors' efforts and hard work and I encourage my fellow reviewers to also reconsider their scores based on the authors' constructive responses and improvements.

---

### Official Review · Reviewer_i5Cv · 2024-11-03

**Soundness:** 3
**Presentation:** 3
**Contribution:** 2
**Rating:** 5
**Confidence:** 2

**Summary:**

This paper proposes an improvement to the representation engineering (RepE) analysis of LLMs. In the original RepE, different treatments are applied to the LLM inputs, and the resulting difference in the activation is taken as the effect of the treatment. This paper observes that the treatment is sometimes unsuccessful, resulting in unfaithful explanations. Thus, this paper proposes to use an external mechanism, such as a content moderation tool for safety analysis, to filter out inputs that do not lead to the targeted behavior change. Experimental analysis showing that this simple modification improves the RepE analysis quality.

**Strengths:**

This paper contains a simple yet effective idea.

The paper is fairly well-written.

Different ways to identify the linear safety templates are used and compared in the experiments.

**Weaknesses:**

While I am not very familiar with the RepE work, in my opinion the proposed modification seems to be incremental and too straightforward. Therefore, given the popularity of the original RepE work, I would defer to other reviewers in assessing whether such proposals have been made before in the literature.

I find the paper writing being unnecessarily complicated by the analogy to neuroscience, and sometimes even the causality machinery. The study of network activations have a long history in interpretability, dating back to (at least) the TCAV work [1]. As such, I am concerned that the neuroscience analogy does not facilitate understanding, but instead could cause further confusion.

For causality, I am not sure if the filtering approach truly leads to the desired causality. Even without filtering (as in the original RepE method), the activation may still have a causal effect on the behavior, just that the effect is not strong enough to be observed. With filtering and pairing, it is possible that the activation and the model behavior difference are both caused by the instruction, so that the activation does not causally explain the behavior difference.

[1] https://arxiv.org/pdf/1711.11279

**Questions:**

In addition to the weaknesses listed above, I have a question about the description in Line 134-136 to create P(Y_x). Doesn't data selection approach create a conditional distribution P(y|x), rather than an interventional distribution? If this creates the interventional distribution, what is the operation that creates the conditional distribution?

---

> ### Author Response · Authors · 2024-11-24
> **Response to Reviewer i5Cv**
>
> Thank you very much for taking the time to review our work and for providing valuable feedback.
> > The proposed modification seems to be incremental and too straightforward.
>
> We respectfully disagree with the characterization of our approach as incremental. As noted by Reviewer 1, the issue we address has not been previously explored. While the proposed solution is straightforward, identifying the vulnerability in representation engineering and designing a principled way to address it is both non-trivial and impactful. Given the widespread use of representation engineering for understanding and improving model behaviors (especially safety), ensuring its reliability is essential for achieving these goals.
>
> The simplicity of our method makes it an ideal first step toward addressing the reliability of representation engineering. The existing body of work focuses on applying RepE in various contexts, but few studies examine its underlying assumptions or its reliability. Our contribution fills this gap with a novel and practically effective approach that lays the groundwork for advancing this research area. Thus, while simple, our method is not incremental but foundational.
>
> > The analogy to neuroscience may be unnecessarily complicated.
>
> We appreciate your feedback on this point and have streamlined these discussions in the revised manuscript. The analogy is chosen to highlight an important parallel: much like in early neuroimaging studies, where researchers attempted to infer human behavior (e.g., honesty or criminality) from brain scans, we find that current representation engineering researches face a similar risk. Without establishing a reliable causal relationship between neural activities and behaviors, the interpretations derived from these methods can be misleading or even detrimental if used for safety-relevant applications. By drawing on lessons from cognitive neuroscience, we hope to emphasize the importance of causally grounded analyses in LLM interpretability. Therefore, we streamline some discussions while retaining key references for readers who may find these parallels insightful and wish to draw inspiration from them.
>
> > Even without filtering (as in the original RepE method), the activation may still have a causal effect on the behavior ...
>
> We appreciate the opportunity to clarify this point. As you noted, "may still have" means that the identified activation does not consistently cause the intended model behavior. The absence of reliable causal effects reduces the utility of RepE in explaining or controlling behaviors. Our method, built on controlled experiments and supported by extensive empirical results, helps isolate causal effects effectively, thus improving the reliability of representation engineering.

---

> > ### Author Response · Authors · 2024-11-24
> > **Response to Reviewer i5Cv (cont.)**
> >
> > > The differences between interventional vs. conditional distribution. Doesn't data selection approach create a conditional distribution rather than an interventional distribution?
> >
> > Interventional and conditional distributions are fundamental yet often misunderstood concepts in causal inference. A conditional distribution, $P(Y | X = x)$, is **the subset of a population** where a condition $X = x$ is observed. In contrast, an interventional distribution, $P(Y_x)$, is obtained by intervening in the system to set the variable $X$ to a specific value $x$ across the entire population, thereby creating **a new population**. If we could intervene to create two populations that are identical except for the value of $X$, then the difference in outcomes $Y$ can be causally attributed to $X$.
> >
> > In CARE, the data selection process approximates interventional distributions by constructing two groups with carefully controlled neural activities $X$. Specifically:
> > - One group contains safe-inducing neural activities $x_1$.
> > - The other group contains risk-inducing neural activities $x_2$.
> >
> > These groups simulate $P_{X=x_1}(Y)$ and $P_{X=x_2}(Y)$, as each pair of samples differs only in their neural activities while holding other factors constant. This ensures that the observed differences in model behavior $Y$ can be causally attributed to the change in neural activity $X$.
> >
> > In representation engineering, both understanding and controlling model behaviors rely on establishing a causal relationship between neural activities and model behaviors. However, the original RepE paper does not address this need. Instead, its effectiveness heavily depends on an implicit assumption: assigning roles in the prompt creates the two interventional distributions mentioned above. In practice, the collected data neither forms true interventional distributions nor conditional distributions with respect to $X$ and $Y$. This lack of clarity undermines the reliability of RepE's conclusions.
> >
> > Thank you for raising this question. As Judea Pearl has noted, the lack of a mathematical language for causality hinders scientific progress. By formalizing and analyzing these causal relationships, we take an important step toward improving the reliability of representation engineering.

---

> > > ### Comment · Reviewer_i5Cv · 2024-11-26
> > >
> > > Thank you for answering my questions. However, my concerns still remain. For my question on "Even without filtering (as in the original RepE method), the activation may still have a causal effect on the behavior", my use of "may still have" represents a possible causal relationship between the instruction and the activation, even if the generated response does not apparently instruction-follow. In this case, is it right to do the filtering as proposed?
> > >
> > > For the interventional vs conditional distribution, I am still confused. In your case, $X$ is the neural activity, either safe-inducing or risk-inducing. However, the $X$ is not directly being intervened upon. Instead, you use instructions to elicit hopefully safe vs. risky behaviors, and then discard data that do not exhibit the desired behaviors, using a content moderation model. This still looks like creating a conditional distribution to me.
> > >
> > > In addition, I share the concerns of other reviewers on the limited novelty, and would like to maintain my score.

---

> > > > ### Author Response · Authors · 2024-11-27
> > > > **Response to Reviewer i5Cv**
> > > >
> > > > Thank you for clarifying your question. We address your concerns below:
> > > >
> > > > > In this case, is it right to do the filtering as proposed?
> > > >
> > > > To address this, we draw a parallel with controlled trials in medicine to explain why filtering (and the broader goal of matching) is essential in our context.
> > > >
> > > > In a well-designed trial, participants are assigned to a treatment or control group to eliminate confounders, ensuring that any observed effect is attributable to the treatment itself. For example, consider a drug efficacy study where patients in the treatment group are expected to take the prescribed medication. If some patients in the treatment group fail to take the medication, their data would introduce noise, making it harder to isolate the causal effect of the drug. In such cases, a common practice is to analyze only the data from patients who adhered to the treatment protocol, filtering non-adherent cases to ensure the validity of the causal inference.
> > > >
> > > > Similarly, in our work, filtering out cases where the model does not follow instructions ensures that observed differences in behavior are causally linked to differences in neural activities. Without filtering, the data would contain confounding factors that obscure the causal relationship between neural activities and model behavior.
> > > >
> > > > > For the interventional vs conditional distribution, I am still confused.
> > > >
> > > >  To further clarify, using the medical example again, constructing treatment and control groups in controlled trials often relies on matching participants based on confounding variables (e.g., age, gender, pre-existing conditions). Although we cannot intervene directly on participants' biology in reality, matching allows us to simulate the effects of interventions as if they were applied to identical populations.
> > > >
> > > > In our work, while we do not directly intervene on neural activities, we achieve a similar goal through matching. By constructing pairs of data points that differ only in their neural activity type (safe-inducing vs. risk-inducing) while holding other factors constant, we approximate the interventional distributions $P(Y_x)$. This method ensures that differences in outcomes $Y$ can be attributed to the neural activity $X$, rather than to confounding factors.
> > > >
> > > > Therefore, while the process may resemble conditioning, its design and intent align with the principles of constructing interventional distributions, as it isolates the causal effect of $X$ on $Y$.  We hope this addresses your concern.
> > > >
> > > > > Novelty
> > > >
> > > > We believe there is a significant misunderstanding regarding the novelty of our work. We have provided a detailed response to this concern in a separate reply. We kindly encourage you to review it and consider the arguments presented. Thank you very much for taking the time to provide valuable feedback.

---

> > > > ### Author Response · Authors · 2024-12-02
> > > >
> > > > Thank you again for your thoughtful review. As we approach the end of the discussion phase, we kindly ask if our responses have addressed your concerns.
> > > >
> > > > If so, we would be grateful if this could be reflected in the review scores. Best regards.

---

### Official Review · Reviewer_givs · 2024-11-04

**Soundness:** 3
**Presentation:** 3
**Contribution:** 3
**Rating:** 5
**Confidence:** 4

**Summary:**

The paper proposes a framework to isolate the impact of confunding factors on correlations between neural activities and model behaviors. The identification of spurious correlations makes causal relation between neural activities and model behaviors more reliable and accurate.

**Strengths:**

- This paper has focused on an interesting problem in LLM interpretability, as the spurious correlations between neural activities and model behaviors is not fully explored yet.
- The proposed framework adopts the philosophy of matched-pair trial design, which helps to filter out the data involved in confunding factors.
- The experiments are comprehensive to demonstrate the effectiveness of the proposed method.

**Weaknesses:**

- The "spurious correlations" in safety-relevant biases could also caused by alignment fine-tuning. It would be more persuasive if the authors can provide more evidence to show the existence of spurious correlations in other cases.
- The failure of instruction following can hardly be treated to confounding factors as mentioned before.

**Questions:**

- What is the percentage of data with confunding factors when evaluating baseline methods?
- Could you please explain why LR is the worst performing model in baseline experiments in terms of manipulation score? Since it usually performs better in intervention tasks compare to mean differences. Is it because of too few data points i.e. stimulus pairs?
- Typo in Table 3, "Manipuation Score".

---

> ### Author Response · Authors · 2024-11-24
> **Response to Reviewer givs**
>
> We would like to express our sincere gratitude for your detailed feedback and insightful questions. Below, we address each of your concerns in details.
>
> > The link between failures of instruction-following and spurious correlation.
>
> Thank you for raising this point.  In the context of this paper, our focus is on cases where inconsistent behavior during neural activity collection leads to spurious correlations. When data includes such inconsistencies, the identified neural activities may not be the cause of model behaviors (e.g., safe and unsafe), thus compromising the reliability of representation engineering.
>
> >  Safety-relevant biases could also caused by alignment fine-tuning.
>
> To address your concern, we conducted experiments on uncensored models, i.e., models trained on datasets where responses containing alignment / moralizing were removed, thereby the model is not biased towards alignment. The results confirm that the existence of spurious correlations leads to unreliable representation engineering, even in the absence of alignment fine-tuning.
>
> | Model | Manipulation Score (safe) | Manipulation Score (unsafe) | Termination Score (safe) | Termination Score (unsafe) | Accuracy     | Precision    | TPR          | TNR          |
> | ----- | ------------------------- | --------------------------- | ------------------------ | -------------------------- | ------------ | ------------ | ------------ | ------------ |
> | BASE  | 40.57 ± 2.38              | 76.04 ± 2.38                | 65.28 ± 1.30             | 73.33 ± 0.83               | 75.15 ± 0.00 | 97.08 ± 0.00 | 67.48 ± 0.00 | 94.79 ± 0.00 |
> | PAIR  | 40.57 ± 2.38              | 76.04 ± 2.38                | 65.28 ± 1.30             | 73.33 ± 0.83               | 75.15 ± 0.00 | 97.08 ± 0.00 | 67.48 ± 0.00 | 94.79 ± 0.00 |
> | CARE  | 98.46 ± 0.54              | 100.00 ± 0.00               | 69.84 ± 0.94             | 78.75 ± 0.83               | 74.99 ± 0.03 | 97.16 ± 0.05 | 67.20 ± 0.08 | 94.97 ± 0.09 |
>
> We focused on safety-related topics because it is a critical application of representation engineering. Using existing content moderation models allowed us to efficiently label data without the need for recruiting and training human labelers, making it valuable for the research community to verity the vulnerability of representation engineering with limited computing resources.
>
> > Percentage of data with confounding factors for baseline.
>
> About 50% of samples showed behavior inconsistent with instructions for baseline.
>
> > Why LR performs worse than mean differences in baseline?
>
> The underperformance of LR aligns with the results reported in the original RepE paper, where LR was the weakest method. The authors of the original RepE paper recommended PCA as the preferred approach but did not provide further explanations for LR's poor performance. Our experiments show that when confounding factors are properly controlled, LR achieves high manipulation scores, suggesting its earlier poor performance was due to sensitivity to confounders rather than inherent limitations of the algorithm.
>
> > Typo in Table 3, "Manipuation Score".
>
> Thank you for pointing out this typo. We have corrected it in the revised manuscript.

---

> > ### Comment · Reviewer_givs · 2024-11-25
> >
> > Thank you for the detailed response. I haven't found solid evidence supporting "inconsistent behavior during neural activity collection leads to spurious correlations" in the rebuttal.
> >
> > I believe there is a misunderstanding about my comment "The 'spurious correlations' in safety-relevant biases could also caused by alignment fine-tuning". This suggests that the safety-relevant biases in this paper could result from the models used not being well-aligned enough. To resolve this, experiments on better-aligned models are needed. While additional experiments on larger LLMs were provided, the results seems confirmed my concern.
> >
> > I also agree with the reviewer frYB's comment on the paper's novelty. So I will keep my score unchanged.

---

> > > ### Author Response · Authors · 2024-11-27
> > > **Response to Reviewer givs**
> > >
> > > > Experiments on better-aligned models are needed.
> > >
> > > Thank you for clarifying your concern. We greatly value your feedback and have conducted additional experiments to address this concern.
> > >
> > > Specifically, we conducted experiments using a state-of-the-art, highly aligned model: LLaMA 3.1-8B. The results are as follows:
> > >
> > > | Model | Manipulation Score (safe) | Manipulation Score (unsafe) | Termination Score (safe) | Termination Score (unsafe) | Accuracy     | Precision    | TPR          | TNR          |
> > > | ----- | ------------------------- | --------------------------- | ------------------------ | -------------------------- | ------------ | ------------ | ------------ | ------------ |
> > > | BASE  | 30.20 ± 4.03              | 21.27 ± 1.23                | 87.18 ± 0.27             | 66.73 ± 1.59               | 30.96 ± 2.09 | 94.14 ± 0.91 | 19.52 ± 2.89 | 92.90 ± 2.27 |
> > > | CARE  | 54.80 ± 5.20              | 97.82 ± 1.87                | 88.72 ± 0.29             | 66.18 ± 1.06               | 62.26 ± 1.61 | 95.12 ± 0.50 | 58.36 ± 2.36 | 83.42 ± 2.57 |
> > >
> > > These results show that even with better-aligned models, the vulnerabilities of representation engineering persist, compromising its reliability. We hope this addresses your concerns.
> > >
> > > > Novelty
> > >
> > > We believe there is a significant misunderstanding regarding the contribution of our work. We have provided a detailed response to this concern in a separate reply. We kindly encourage you to review it and consider the arguments presented. Thank you very much for taking the time to provide valuable feedback.

---

> > > ### Author Response · Authors · 2024-12-02
> > >
> > > Thank you again for your thoughtful review. As we approach the end of the discussion phase, we kindly ask if our responses have addressed your concerns.
> > >
> > > If so, we would be grateful if this could be reflected in the review scores. Best regards.

---

### Author Response · Authors · 2024-11-25
**Summary of Changes**

We sincerely thank all reviewers for their thoughtful feedback, which has been invaluable in refining and strengthening our work.

The reviewers recognized the strength of our experiments, noting their rigor and comprehensiveness. Reviewer givs highlighted them “comprehensive to demonstrate the effectiveness of the proposed method,” and Reviewer G4PV highlighted the clear improvements shown over baselines. These acknowledgments validate our effort to ensure experimental robustness, including running each experiment five times with different random seeds and reporting interval estimates to **reach reliable conclusions**.

Our paper’s motivation and practical significance were also widely appreciated. Reviewers givs, i5CV, and G4PV agreed that our work addresses a "genuine limitation in RepE" (Reviewer G4PV) and focuses on an "interesting problem in LLM interpretability" (Reviewer givs). Additionally, Reviewer frYB noted that our proposed approach is "principled, easy to implement, and useful for practitioners. This feedback highlights the significance of our contributions, **not only in proposing a novel method but also in identifying a key research question that worth further investigation**.

Additionally, the clarity of the writing received positive feedback. Reviewer i5Cv described the paper as "fairly well-written," and Reviewer frYB found it "easy to follow." These comments motivate our continued effort to **present our work in a clear and accessible manner**.

**Summary of Changes**

Based on the constructive feedback, we made the following changes:
- The original Figure 1, which presented a neuroscience analogy, has been moved to Appendix A.1, reducing the cognitive load for readers, as suggested by Reviewer i5Cv.
- A new Figure 1 has been added to the introduction, providing a concrete example of inconsistent model behavior, as suggested by Reviewer frYB.
- We have included “evaluation circularity” as a future direction in the conclusion, as suggested by Reviewer G4PV. This section highlights the potential issues arising from dependencies between filtering mechanisms and evaluation methods and proposes avenues for mitigating these in future work.
- We conducted additional experiments using ShieldGemma-9b as an alternative moderation model. The results, included in Appendix A.6, demonstrate that the conclusion is consistent across different moderation tools.
- We conducted additional experiments using both smaller (1.1B) and larger (13B) LLMs to evaluate scalability. These results, included in Appendix A.7, confirm that our method performs reliably across different model sizes, and point to promising directions for future research.

These changes address the key concerns raised by the reviewers and further enhance the clarity of our work. As the first step in exploring the reliability of representation engineering, this paper identifies a key limitation and proposes a principled  approach that lays the foundation for future research. We are grateful for the opportunity to incorporate your feedback and hope that our revisions meet your expectations. Thank you!

---

> ### Comment · Reviewer_frYB · 2024-11-25
> **Respectfully disagree with this characterisation**
>
> I wish to disagree with the characterisation of some parts of this summary, specifically focusing on ones where I am quoted.
>
> > Additionally, Reviewer frYB noted that our proposed approach is "principled, easy to implement, and useful for practitioners. This feedback highlights the significance of our contributions, not only in proposing a novel method but also in identifying a key research question that worth further investigation.
>
> The "proposed approach" is an incremental change (filtering responses where the model doesn't follow instruction) on the main LAT pipeline in the RepE paper, and hence not novel, significant, or a key research question worth further investigation in my opinion. This opinion on this can differ, but it is important to point out I disagree with this claim made by quoting parts of my review.
>
> > We conducted additional experiments using both smaller (1.1B) and larger (13B) LLMs to evaluate scalability. These results, included in Appendix A.7, confirm that our method performs reliably across different model sizes, and point to promising directions for future research.
>
> I had asked for these experiments as I thought the effectiveness of the proposed change to the LAT pipeline would decrease as instruction following capabilities improve. The results in Appendix A.7 only confirm this, and I would not use them to claim the method performs reliably across different model sizes. In fact, I believe if this approach is used on state of the art models, the improvement will be negligible, but this isn't tested in the paper.

---

> > ### Author Response · Authors · 2024-11-27
> > **Response to Reviewer frYB**
> >
> > Thank you for clarifying your perspective. We value your feedback as an opportunity to seek consensus and resolve concerns to improve our work.
> >
> > > The "proposed approach" is an incremental change ... and hence not novel, significant, or a key research question worth further investigation in my opinion.
> >
> > We must clarify that the contribution of this work goes far beyond “an incremental change.” CARE establishes a principled framework for addressing a critical vulnerability in representation engineering: the implicit assumption of consistent behavior during neural activity collection. This is supported by extensive empirical results. Suggesting the existing method is already reliable and does not warrant further investigation would be overly optimistic and inconsistent with the empirical findings presented in the paper. We will provide a separate reply for the concerns on novelty.
> >
> > > The results in Appendix A.7 only confirm that the effectiveness of the proposed change decreases as instruction-following capabilities improve.
> >
> > We interpret these results differently. Specifically:
> >
> > 1. CARE consistently improves performance across all model sizes: For the 13B model, the manipulation score (safe) is 21.75 ± 0.81 vs. 18.81 ± 1.39. Similarly, the manipulation score (unsafe) is 49.33 ± 3.03 vs. 45.56 ± 1.99. These scores still show an improvement, so we don't interpret them as negative results.
> > 2. Even if these results are criticized for the low success rate of manipulation, they still demonstrate the importance of investigating the reliability of representation engineering, not the opposite. We genuinely thank you for raising this point, as it highlights an interesting direction for future research.
> >
> > > If this approach is used on state-of-the-art models, the improvement will be negligible.
> >
> > CHALLENGE ACCEPTED! We took this challenge seriously. The main model we use throughout the paper is LLaMA 3. We conducted additional experiments on LLaMA 3.1, a state-of-the-art model with advanced instruction-following capabilities. This model has not been fine-tuned on uncensored datasets. The new results, included in Appendix A.8, show that CARE consistently improves over BASE (54.80±5.20 vs. 30.20±4.03 for safe instructions, 97.82±1.87±21.27±1.23 for unsafe instructions), suggesting CARE remains effective for state-of-the-art models.
> >
> > We hope these results address your concerns. If you think LLaMA 3.1 is still not SoTA enough and would like to see additional experiments for other SoTA models (<13B since we only have limited resources), please let us know. We will try our best to address your concerns.

---

> > > ### Author Response · Authors · 2024-11-27
> > > **Response to Concerns on Novelty**
> > >
> > > According to the Reviewer Guide, novelty is defined as whether the paper presents novel findings. We strongly believe that our work meets this criterion for the following reasons:
> > >
> > > **Addressing an unexplored problem**: As of November 27, 2024, Google Scholar lists 226 works citing the RepE paper, discussing its applications in hallucination mitigation, representation-based alignment, adversarial attacks, jailbreaking, and knowledge manipulation. However, 0 papers study the reliability of representation engineering itself. Therefore, the problem being studied is unexplored, and our work serves as a first step to address it.
> > >
> > > **Filling methodological gaps in RepE**: This contribution aligns with Reviewer frYB’s acknowledgment: “This work does a great job of rigorously analysing ..., which has higher importance because this is done poorly in the original RepE paper.” We clarify and formalize the goals of representation engineering using causal science terminology. For the first time, we explicitly articulate that RepE seeks to establish causal relationships between neural activities and model behaviors. This clarification reveals the implicit assumptions in the original RepE methodology and highlights the potential consequences when these assumptions are violated. Drawing a parallel with cognitive neuroscience, the evolution from identifying superficial correlations to uncovering causal mechanisms has been a transformative step in establishing rigorous scientific disciplines.
> > >
> > > **Extensive experimental validation**：To substantiate our claims, we conducted a large number of experiments spanning five safety-relevant aspects. These experiments not only demonstrate the practical limitations of existing RepE methodologies but also show that the proposed CARE framework significantly improves reliability. As shown in Figures 5 and 6, CARE achieves robust performance improvements with much narrower confidence bands compared to the baselines, providing a principled approach to improve  RepE's applicability.
> > >
> > > Based on these points, we respectfully disagree with reducing these efforts to “just filtering responses,” as this overlooks the depth and significance of our findings. The originality of our work lies in identifying when representation engineering fails, understanding the cause of these failures, explaining why filtering can mitigate the issue, and proposing a principled implementation of filtering within a causal framework. These are fundamental questions that have not been addressed in prior works and contribute to providing **novel findings** in the field of representation engineering.
> > >
> > > We sincerely thank all reviewers for their time, and actionable feedback, which has been invaluable in strengthening our work. We hope this response clarifies your concerns.

---

> > > > ### Comment · Reviewer_frYB · 2024-11-28
> > > >
> > > > > Suggesting the existing method is already reliable
> > > >
> > > > This was not suggested. Instead, I meant that the specific contribution made by this paper, filtering responses, provides diminishing results for models better at instruction following. The models studied in this paper are bad at instruction following, which is why gains are observed. I agree it is useful to know for practitioners that small models which are bad at instruction following should include this filtering step when applying RepE.
> > > >
> > > > As for the results provided for Llama 3.1 8B, I appreciate the authors running their method on this model, but like most results in the paper, these are not proposed relative to the LAT method in the RepE paper, which is what this paper builds on. I believe most of the gains already come from the LAT methodology adopted, and less so from the filtering added on top by the proposed method CARE. Could you please report LAT results alongside, and also what fraction of stimuli were filtered for Llama 3.1 8B by your methodology?
> > > >
> > > > As for calling this contribution a "foundational methodological change", an "interesting direction for future research", "principled framework" etc. This is up for value judgement, but in my opinion this is over claiming and merely wrapping a story around a small contribution. Like I said, this opinion can differ, but the posted summary misrepresented my opinion as the author's opinion, so it was important for me to clarify.

---

> > > > > ### Author Response · Authors · 2024-12-02
> > > > > **Further Responses (1/2)**
> > > > >
> > > > > Thank you very much for reading our responses and taking the time to provide further feedback. We have reached some common ground, and we appreciate the opportunity to address the remaining concerns.
> > > > >
> > > > > > The models studied in this paper are bad at instruction following, which is why gains are observed.
> > > > >
> > > > > We believe this statement misrepresents our work and may lead to **misunderstandings**. To provide additional context, here are some recent works on representation engineering and the models they study:
> > > > > - Zhang et al. (NeurIPS 2024) [1] use LLaMA 2-7B, Vicuna-7B, and Guanaco-7B (P7, Sec. 5.1).
> > > > > - Zou et al. (NeurIPS 2024) [2] use Mistral-7B and LLaMA 3-8B (P6, Sec. 4.1).
> > > > > - Anonymous authors ([3]) (in submission to ICLR 2025) use LLaMA 2-7B and other 7B models (P6, Sec. 5.2.2).
> > > > > - Anonymous authors ([4]) (in submission to ICLR 2025) use LLaMA 3-8B, Mistral-7B, and Gemma-7B (P8, Sec. 5).
> > > > >
> > > > > When designing our experiments, we chose LLaMA 3-8B because it was one of the strongest instruction-following models of its size at that time. Labeling the models as “bad at instruction following” does not accurately reflect the current state of the field and undermines the efforts we made. While we agree that exploring more advanced models is an important direction, current research mainly focuses on medium-sized models, making our findings broadly relevant.
> > > > >
> > > > > > Like most results in the paper, these are not proposed relative to the LAT method in the RepE paper.
> > > > >
> > > > > This comment is based on a **misunderstanding**, which we have addressed in a separate response. To ensure clarity for readers of this thread, we provide additional context.
> > > > >
> > > > > In the original RepE paper and its open-source implementation, there are two approaches to constructing the stimulus set, which we refer to as **BASE** and **PAIR** in our paper:
> > > > > - **BASE**: Assigns different roles to each stimulus pair without requiring text stimuli pairs to be comparable in content (see the Honesty experiment in the open-source code).
> > > > > - **PAIR**: Assigns different roles while ensuring text stimuli are paired and comparable (see the Harmfulness experiment). This is also referred to as "algorithm 1 line 7 and 8", "the LAT method" or "essentially the same as the matched-pair design" in Reviewer frYB's comments.
> > > > >
> > > > > Neither of these methods satisfies the criteria for “matched-pair trial design,” as they fail to account for potential confounding between the neural activities and the model behaviors, and do not control for confounding factors.
> > > > >
> > > > > Our paper presents extensive comparisons against both baselines. All experiments were conducted five times with different random seeds to account for uncertainty and we additionally report confidence intervals estimated using percentile bootstrap with stratified sampling. Therefore, we respectfully disagree with this characterization.
> > > > >
> > > > > > I believe most of the gains already come from the LAT methodology adopted. Could you please report LAT results alongside, and also what fraction of stimuli were filtered for LLaMA 3.1 8B by your methodology?
> > > > >
> > > > > We are happy to provide additional insights, while also highlighting the original results presented in the paper should not be overlooked.
> > > > > - In experiments with LLaMA 3-8B (uncensored), the manipulation scores (safe) for CARE vs. PAIR were 93.84 ± 0.94 vs. 57.21 ± 7.25, and manipulation scores (unsafe) were 99.47 ± 0.38 vs. 58.71 ± 17.06.
> > > > > - In new experiments with LLaMA 3.1-8B (censored), the manipulation scores (safe) for CARE vs. PAIR were 54.80 ± 5.20 vs. 48.99 ± 2.28, and manipulation scores (unsafe) were 97.82 ± 1.87 vs. 77.45 ± 12.93. About 87% of stimulus pairs exhibited inconsistent behaviors, mainly due to the model refusing to generate unsafe responses.
> > > > >
> > > > > For censored models, we observed that manipulation scores (safe) decreased across all baselines, indicating that suppressing neural activity to generate unsafe responses becomes harder. Additionally, the relative advantage of PAIR over BASE becomes larger, but CARE still yields the best results.

---

> > > > > > ### Author Response · Authors · 2024-12-02
> > > > > > **Further Responses (2/2)**
> > > > > >
> > > > > > > In my opinion this is over claiming and merely wrapping a story around a small contribution.
> > > > > >
> > > > > > We acknowledge that terms like "fundamental change" are subject to value judgement. It was used in response to Reviewer frYB's comment "this isn't a fundamental change in design." and was not intended to overclaim but rather to emphasize the importance of the findings. We avoided such subjective terms in the paper. We encourage readers to refer to the stated contributions (P2–P3), which are carefully stated, evidence-based, and presented with appropriate caution. Furthermore, we explicitly acknowledges the limitations of our work in the conclusion (P10).
> > > > > >
> > > > > > We disagree with the characterization of our work as "merely wrapping a story around a small contribution", which overlooks the substantial effort invested in identifying the vulnerability in representation engineering, proposing a principled solution, and empirically validating its effectiveness across diverse tasks.
> > > > > >
> > > > > > Representation engineering, inspired by cognitive neuroscience, is becoming increasingly popular as a way to explain and control model behaviors via neural activities. At the same time, similar methods in cognitive neuroscience have long struggled with issues of correlation vs. causation ([5]). Our work demonstrates, through extensive experiments, that concerns about the reliability of representation engineering are real and merit attention. These findings are not "merely wrapping a story" but highlight a previously unexplored vulnerability in this growing field.
> > > > > >
> > > > > > We really appreciate Reviewer frYB's feedback and clarifications. We hope the discussion helps address concerns and provides a more comprehensive view of our work. Thank you!
> > > > > >
> > > > > > [1] Zhang et al., "[Adversarial Representation Engineering: A General Model Editing Framework for Large Language Models](https://openreview.net/forum?id=dQ9ji8e9qQ)." NeurIPS 2024.
> > > > > >
> > > > > > [2] Zou et al., "[Improving Alignment and Robustness with Circuit Breakers](https://openreview.net/forum?id=IbIB8SBKFV)." NeurIPS 2024.
> > > > > >
> > > > > > [3] Anonymous., ”[REEF: Representation Encoding Fingerprints for Large Language Models](https://openreview.net/forum?id=SnDmPkOJ0T)," in submission to ICLR 2025.
> > > > > >
> > > > > > [4] Anonymous., ”[Robust LLM safeguarding via refusal feature adversarial training](https://openreview.net/forum?id=s5orchdb33)," in submission to ICLR 2025.
> > > > > >
> > > > > > [5] Siddiqi, Shan H., et al. "[Causal mapping of human brain function](https://www.nature.com/articles/s41583-022-00583-8)." Nature reviews neuroscience 23.6 (2022): 361-375.

---

> ### Comment · Reviewer_frYB · 2024-12-03
>
> >  and do not control for confounding factors.
>
> Could you please list what these confounding factors are, and quantify the fraction of stimuli each of the confounders affect?

---

> > ### Author Response · Authors · 2024-12-04
> >
> > > Could you please list what these confounding factors are, and quantify the fraction of stimuli each of the confounders affect?
> >
> > Thank you for raising this question. In the original submission, we explained this concept using causal graphs in Figure 1. Following Reviewer i5Cv's feedback, we moved this discussion to Appendix A.1 (Figure 8) for reduced cognitive load.
> >
> > In causal inference, confounding factors are variables that influence both the treatment and the outcome, making observed associations fail to reliably reflect causal relationships. In the context of representation engineering, understanding this concept can be challenging, so we drew an analogy to neuroscience. As shown in Figure 8, confounding factors like "stress" might influence both measured brain activity and observed human behaviors, thereby making it difficult to reliably attribute behaviors to brain activity.
> >
> > In representation engineering, we avoid anthropomorphic terms like "stress" to prevent debates about whether models exhibit human-like emotions or experiences. Instead, we use "instruction", which influence both the neural activities identified by representation engineering and the model behaviors. For example, if the model refuses to follow unsafe instructions, the identified neural activities might not accurately indicate the direction for unsafe behaviors, introducing confounding bias and reducing the effectiveness of representation engineering, as shown in our experiments.
> >
> > The proportion of stimuli affected by confounding factors varies with the model and dataset. For instance, in experiments with LLaMA 3.1-8B on the Weapons & Regulated Substances dataset, approximately 87% of stimulus pairs were affected due to inconsistent instruction-following behavior.

---

### Meta-Review · Area_Chair_UXiG · 2024-12-20

**Metareview:**

This paper proposes a modification to representation engineering to improve the reliability of the technique. The reviewers largely agreed that the paper's idea was clear and ably executed. However, there was also a sense that the scope of the contribution is not adequate to represent significant progress. Unfortunately, this has been framed in terms of novelty of the contribution, which is a poor measure of quality. However, the point remains that the paper does not argue for the significance of its own contributions in a manner adequate to convince the reviewers. Thus, the paper requires either a major revision to the exposition---making it clear what the importance of the work is---or some further extension. As such, it is not yet ready for publication at ICLR.

**Additional Comments On Reviewer Discussion:**

see above

---

### Decision · Program_Chairs · 2025-01-22

Reject